Contribution of metabolomics to the taxonomy and systematics of octocorals from the Tropical Eastern Pacific

http://orcid.org/0000-0002-2547-4152 Jaramillo Karla B. 1 2 3 4 kbjarami@espol.edu.ec
http://orcid.org/0000-0001-8270-5562 Guillén Paúl O. 1 4 5
http://orcid.org/0000-0003-1546-2307 Abad Rubén 4
Rodríguez León Jenny Antonia 3 4
McCormack Grace 2
1 Marine Biodiscovery, School of Chemistry and Ryan Institute, National University of Ireland, Galway , Galway , Ireland
2 Zoology, School of Natural Sciences and Ryan Institute, National University of Ireland, Galway , Galway , Ireland
3 Facultad de Ciencias de la Vida, Escuela Superior Politécnica del Litoral , Guayaquil , Ecuador
4 Centro Nacional de Acuicultura e Investigaciones Marinas, CENAIM, Escuela Superior Politécnica del Litoral , Guayaquil , Ecuador
5 Facultad de Ciencias Naturales y Matemáticas, Escuela Superior Politécnica del Litoral , Guayaquil , Ecuador
Banaszak Anastazia
Electronic publication date: 2025 Mar 12
Publication date: 2025
Volume: 13
Electronic Location ID: e19009
Received 2024 Feb 9; Accepted 2025 Jan 27
Copyright: © 2025 Jaramillo et al.
Copyright year: 2025
Copyright holder: Jaramillo et al.
License: This is an open access article distributed under the terms of the Creative Commons Attribution License, which permits unrestricted use, distribution, reproduction and adaptation in any medium and for any purpose provided that it is properly attributed. For attribution, the original author(s), title, publication source (PeerJ) and either DOI or URL of the article must be cited.
License URL: https://creativecommons.org/licenses/by/4.0/

Keywords: Ecuador, Chemotaxonomy, Phylogenetic, Biodiversity, Cnidaria, Invertebrate, DNA, Metabolites, Secondary metabolites, Corals

Funding: Secretaría de Educación, Ciencia y Tecnología e Innovación of Ecuador (SENESCYT) PIC-14-CENAIM-001 ESPOL University of Ecuador CENAIM-400-2019 Marine Institute PBA/MB/16/01 Marine Research Programme by the Irish Government NUI Galway This work has been funded by the Secretaría de Educación, Ciencia y Tecnología e Innovación of Ecuador (SENESCYT) in the framework of the PIC-14-CENAIM-001 Project “Characterisation of the Microbiological and Invertebrate Biodiversity of the El Pelado Marine Reserve at Taxonomic, Metabolomic and Metagenomics scales for use in Human and Animal Health”, by the ESPOL university of Ecuador in the framework of the CENAIM-400-2019 Project “Valorization of the Biodiversity through Biodiscovery for its sustainable use in human health and Aquaculture”. Also, the Grant-Aid Agreement No. PBA/MB/16/01 is carried out with the support of the Marine Institute and is funded under the Marine Research Programme by the Irish Government. Finally, NUI Galway for funding Karla B. Jaramillo’s and Paúl O. Guillen’s Ph.D. scholarships. The funders had no role in study design, data collection and analysis, decision to publish, or preparation of the manuscript.

==============================
Octocorals are sessile invertebrates that play a key role in marine habitats, with significant diversity in the Tropical Eastern Pacific, especially in Ecuador’s shallow waters. This study focuses on the most representative octocorals within the Marine Protected Area El Pelado, Santa Elena, Ecuador, as a part of a marine biodiscovery project employing an integrative approach. While molecular techniques have advanced, challenges persist in distinguishing closely related species. Octocorals produce a wide range of compounds, characterized by unique chemical structures and diverse biological properties. Therefore, the main objective of this study was to assess the potential of metabolomics and advanced analytical techniques to analyze the metabolome of these organisms, aiming to refine species classification and improve understanding of octocoral systematics in this region. Untargeted metabolomics effectively discriminates 12 octocoral species across five genera: Muricea, Leptogorgia, Pacifigorgia, Psammogorgia, and Heterogorgia, with notable differentiation between species within the genus Muricea, reinforcing its utility as an additional data set for species characterization. Secondary metabolites such as sterols, steroids, and terpenes (furanocembranolides and sesquiterpenes), were identified in Leptogorgia and Muricea. Overall, this method enabled the identification of 11 known species and a potentially new one, Leptogorgia cf. alba, confirming the extreme diversity of this group in the Tropical Eastern Pacific and within the Ecuadorian marine ecosystem. The study highlights the value of metabolomics in octocoral systematics and encourages for its broader application in marine biodiversity research.

Introduction

‘Animal forests’, settled on rocky reefs and formed by dense communities of sessile marine invertebrates, especially members of the Cnidaria group, represent one of the most important marine ecosystems with a high diversity of microhabits (Jones et al., 2004; Sánchez, 2016; Reverter et al., 2018). Octocorals (Cnidaria: Anthozoa: Octocorallia: Malacalcyonacea) are widespread and abundant in different marine habitats. In addition to stony corals, this group of marine invertebrates has been the focus of several studies (Idjadi & Edmunds, 2006; Plaisance et al., 2011; Sánchez et al., 2019; Stella, Jones & Pratchett, 2010). They are the most abundant and diverse macrofaunal sessile group in temperate and tropical waters (Bayer, 1961; Sánchez et al., 2003; Soler-Hurtado, Megina & López-González, 2018; Velásquez & Sánchez, 2015). Inventories performed in the Tropical Eastern Pacific (TEP) realm reported the order Alcyonacea, now known as Malacalcyonacea, as the most emblematic group in this region (Breedy & Guzman, 2004; Breedy, Guzman & Vargas, 2009), with the presence of endemic genera and species widely distributed in the underwater substrates of mainland coasts, but also around oceanic islands (Abeytia, Guzman & Breedy, 2013; Breedy & Guzman, 2007; Soler-Hurtado et al., 2016).

Recently, several taxonomic studies have provided valuable information to fill gaps in our knowledge of this group of corals across the TEP realm. Approximately 190 species have been reported on the coasts of Ecuador, Colombia, Panama, Costa Rica, and Mexico (Breedy & Cortes, 2008, 2011; Breedy & Guzman, 2004, 2005a, 2005b, 2003a, 2003b; Soler-Hurtado et al., 2016; Steiner et al., 2018; Vergara-Florez et al., 2022). The coast of Ecuador has been recognized as a hotspot of marine biodiversity within the TEP region (Myers et al., 2000), even though its coastline is relatively shorter than those of Chile, Peru, and Colombia. Eight new species of the order Malacalcyonacea have been identified in this region over the past decade (Soler-Hurtado & López-González, 2012; Soler-Hurtado et al., 2016, 2017; Vergara-Florez et al., 2022). Leptogorgia, Muricea, and Pacifigorgia were the most abundant octocoral genera along the coast of Ecuador. They are widely distributed along the coastline and have been described as excellent builders of marine ecosystems (Abad et al., 2022; Soler-Hurtado et al., 2016; Steiner et al., 2018).

Species descriptions of octocorals present in shallow waters (<50 m) of the Eastern Pacific have focused on morphology, particularly sclerite characterization (Breedy & Guzman, 2004, 2005b, 2007, 2003a; Breedy, Guzman & Vargas, 2009; Zeas Valarezo, 2015). Nevertheless, recent studies have highlighted the potential of DNA barcoding for the identification and classification of octocoral species (Sánchez et al., 2003; Soler-Hurtado, López-González & Machordom, 2017; Vargas et al., 2010; Wirshing & Baker, 2015). This combination of morphological features and molecular barcoding sequences has been recognized by the scientific community as an integrative taxonomic method (Dayrat, 2005; Will, Mishler & Wheeler, 2005). Furthermore, other scientific areas, including biochemistry, histology, and ecology, have contributed to the description and classification of species, enabling a better understanding of their origins, geographical boundaries, and historical evolution (Jaramillo, 2019; Ankita et al., 2016; Padial et al., 2010; Singh, 2012). While molecular barcoding sequence proved to be useful in elucidating the phylogenetic relationships of octocoral, for example, by using mt-COI and mt-MutS, the data results could be inconclusive for the species-level discrimination (Vargas et al., 2014). Therefore, the aim of this study was to assess the potential of a more integrative approach, incorporating chemical data, for the species-level separation of octocorals.

The utilization of chemical data for the identification of species has a long historical precedent within the discipline of taxonomy. This has facilitated the generation of invaluable insights into the prospective use of synapomorphic chemical markers for the classification of different taxa, a process currently recognized as chemotaxonomy (Jaramillo, 2019). Secondary or specialized metabolites play a vital role in the marine environment, with some acting as defense mechanisms. These include the production of mucus, stinging substances, and radiant colors to camouflage themselves. The phenotypic traits expressed by these metabolites are vital for the survival of living organisms and display high levels of specificity between species of the same taxon (Croteau, Kutchan & Lewis, 2000; Pichersky & Gang, 2000). In recent decades, metabolomics has emerged as a novel approach to address the limitations of classical chemistry. This is corroborated by sophisticated analytical techniques, including high-resolution mass spectrometry (HR-MS1), computational and statistical analysis of extensive datasets, which are employed to investigate the entire metabolome of a single organism (Patti, Yanes & Siuzdak, 2012; Smith et al., 2006).

Metabolomic datasets have recently been added to integrative systematic studies on different groups of marine invertebrates, such as sponges (Ivanišević et al., 2011), zoantharians (Cachet et al., 2015; Jaramillo et al., 2018), cyanobacteria (Duperron et al., 2020), and soft corals (Aratake et al., 2012; Coll, 1992; Gerhart, 1983; Kessel et al., 2023; Vohsen, Fisher & Baums, 2019), providing valuable insights into their classification. Octocorals are good candidates for the use of metabolomic data, as some species of this group are known to be rich sources of structurally diverse specialized metabolites, also known as natural products, such as sesquiterpenoids, sterols, steroidal glycosides, and saponins, among others (Bandurraga & Fenical, 1985; Berrue & Kerr, 2009; Carroll et al., 2019; Popov, Carlson & Djerassi, 1983). In the context of this study, the octocoral genera Muricea and Leptogorgia have been the subject of the most extensive research in terms of their chemical composition, in comparison to other genera such as Psammogorgia, Heterogorgia, and Pacifigorgia. In fact, chemical studies on Muricea species revealed the presence of sterols (Popov, Carlson & Djerassi, 1983), saponins (Bandurraga & Fenical, 1985; Gutiérrez et al., 2004), steroids and derivatives (Gutiérrez et al., 2006; Lorenzo et al., 2006; Murillo-Álvarez & Encarnación-Dimayuga, 2003; Ortega et al., 2002) while studies on Leptogorgia species reported the presence of polyoxygenated steroids (Cimino et al., 1984; Garrido et al., 2000; Moritz et al., 2014) diterpenes such as cembranolides and furanocembranolides (Dorta et al., 2007; Gutiérrez et al., 2005; Ksebati, Ciereszko & Schmitz, 1984; Missakian, Burreson & Scheuer, 1975; Ortega et al., 2008) halogenated furanocembranolides (Gallardo et al., 2018) sesquiterpenes (Kapustina et al., 2020) and tetraprenylated alkaloids (Keyzers et al., 2006). Chemical studies on Pacifigorgia species have revealed the presence of sesquiterpenes and steroids (Izac et al., 1982a, 1982b; Chen et al., 2016), whereas sesquiterpenes have only been reported in Heterogorgia uatumani (Maia et al., 1999). No chemical studies have been conducted on Psammogorgia species.

Metabolomic studies have been performed on octocoral species to identify bioactive metabolites (Farag et al., 2017; Santacruz et al., 2019; Tanaka, Yoshida & Benayahu, 2005). Santacruz et al. (2019) identified some diterpenoids as specific markers to discriminate the cytotoxic extracts of 28 Caribbean soft corals using a metabolomic approach. Aratake et al. (2012) investigated the relationship between soft coral diversity and cembranoid diterpene production across multiple species of the genus Sarcophyton in Okinawa, Japan using an integrative approach (morphology, phylogeny, and chemotaxonomy). The second metabolomic study focused on characterizing the metabolomic diversity of deep-sea corals in an ecological context by investigating patterns across space and phylogeny (Vohsen, Fisher & Baums, 2019).

In this study, we hypothesized that the metabolomic data of octocorals collected from the Marine Reserve El Pelado (REMAPE) could help in the systematics of this group (Berrue & Kerr, 2009; Kobayashi et al., 1990). Therefore, the aim of our study was to assess the potential of an untargeted metabolomics approach to explore and characterize the different genera of octocorals collected in REMAPE and between the different species of Muricea.

Materials and Methods

Biological material

Octocoral samples were obtained under the research permit N°005-17 IC-FAU-DPSE/Maas, which was granted as part of the framework contract for access to genetic resources “MAE-DNB-CM-2015-0021” by the Ministry of Environment, Water and Ecological Transition of Ecuador (MAATE). In addition, the ESPOL University’s Research Review Board has formally approved the ethical framework for conducting the study in its facilities under the following projects: “Characterization of the invertebrates biodiversity at El Pelado Marine Reserve at Metabolomic Metagenomic Taxonomic scales for use in human and animal health”, ethical application reference: CENAIM-470-3-2014 and “Valorization of the biodiversity through biodiscovery for its sustainable use in human health and Aquaculture”, ethical application reference: CENAIM-400-2019).

A total of 71 octocoral specimens belonging to 12 species were collected during August and September 2017 through SCUBA diving in rocky reef habitats at depths between 10 and 26 m from three different sites (Cuarenta, Pared and Acuario) at the Marine Protected Area “El Pelado” (REMAPE) in the province of Santa Elena, mainland coast of Ecuador as described in Jaramillo et al. (2018) (Fig. 1 and Table 1). For nine species (OC-3, 63, 43, 64, 70, 67, 55, 74, and 20), the same specimen was processed for all three types of approaches: A, a dry subsample was kept for external morphological characterization; B, a subsample was fixed in 95% ethanol for sclerite and molecular analyses and C, a subsample was frozen at –80 °C for metabolomic studies. For the three species, Muricea plantaginea, Muricea squarrosa and Psammogorgia arbuscula, no molecular barcoding sequence could be obtained. However, from three to 12 specimens of each species could be analyzed by metabolomics. Portions of this text have previously been published as part of a thesis (https://researchrepository.universityofgalway.ie/entities/publication/00612b54-607e-411d-bd69-9b09f3cee85f).

Figure 1 In situ images of the 12 Octocorals studied.

(A) Muricea squarrosa. (B) Muricea plantaginea. (C) Muricea austera. (D) Muricea purpurea. (E) Muricea crassa. (F) Muricea fruticosa. (G) Psammongorgia arbuscula. (H) Heterogorgia hickmani (I) Pacifigorgia rubicunda. (J) Leptogorgia obscura. (K) Leptogorgia alba (pink morphotype). (L) Leptogorgia cf. alba (white morphotype). The scale bar represents 1 cm. Photo credit: Karla B. Jaramillo.

Table 1 Octocorals samples.

Species, specimens, localities, voucher numbers and Genbank accession numbers analysed in this study. The sampling sites coordinates can be found in Jaramillo et al. (2018).

Species name	Sample code	Locality	Depth (m)	Morpho	Metabo	COI	MutS1	Voucher	
Muricea fruticosa	OC-1	Acuario	10	_	_	_	_	170816EP02-06	
OC-2	Acuario	25	_	✓	_	_	170816EP02-07	
OC-3	Acuario	10	✓	✓	MN318307	XXXXXXX	170816EP02-08	
OC-4	Acuario	12	_	✓	_	_	170816EP02-09	
OC-33	Cuarenta	22	_	✓	_	_	170816EP07-12	
OC-34	Cuarenta	20	_	✓	_	_	170816EP07-13	
	OC-47	Pared	10	_	✓	_	_	170810EP01-12	
Muricea plantaginea	OC-48	Pared	26	_	✓	_	_	170810EP01-13	
OC-50	Pared	24	_	✓	_	_	170810EP01-15	
OC-51	Pared	25	_	✓	_	_	170810EP01-16	
OC-52	Pared	26	_	✓	_	_	170810EP01-17	
OC-53	Pared	28	✓	✓	_	_	170810EP01-18	
OC-5	Acuario	20	_	✓	_	_	170816EP02-10	
OC-6	Acuario	31	_	✓	_	_	170816EP02-11	
OC-7	Acuario	20	_	✓	_	_	170816EP02-12	
OC-8	Acuario	25	_	✓	_	_	170816EP02-13	
OC-69	Cuarenta	15	_	✓	_	_	170907EP07-14	
OC-25	Cuarenta	10		✓	_	_	170816EP07-04	
Muricea squarrosa	OC-27	Cuarenta	20	_	✓	_	_	170816EP07-06	
OC-28	Cuarenta	31	_	✓	_	_	170816EP07-07	
OC-30	Cuarenta	15	_	✓	_	_	170816EP07-09	
OC-31	Cuarenta	25	✓	✓	_	_	170816EP07-10	
OC-60	Cuarenta	10	_	✓	_	_	170907EP07-05	
OC-61	Cuarenta	10	_	✓	_	_	170907EP07-06	
OC-62	Cuarenta	12	_	✓	_	_	170907EP07-07	
OC-29	Cuarenta	8	_	✓	_	_	170816EP07-08	
OC-57	Cuarenta	10	_	✓	_	_	170907EP07-02	
OC-12	Acuario	10	_	✓	_	_	170816EP02-17	
OC-15	Acuario	15	_	✓	_	_	170816EP02-20	
OC-16	Acuario	12	_	✓	_	_	170816EP02-21	
Muricea crassa	OC-63	Cuarenta	10	✓	✓	MN318310	XXXXXXX	170907EP07-08	
OC-22	Cuarenta	15	_	✓	_	_	170816EP07-01	
OC-56	Cuarenta	12	_	✓	MN318309	XXXXXXX	170816EP07-03	
OC-23	Cuarenta	12	_	✓	_	_	170816EP07-02	
OC-73	Acuario	8	_	✓	_	_	170816EP02-15	
OC-24	Acuario	10	_	✓	_	_	170816EP02-18	
Muricea purpurea	OC-14	Acuario	18	_	✓	_	_	170816EP02-19	
OC-13	Acuario	5	_	✓	_	_	170816EP02-18	
OC-71	Acuario	15	_	✓	_	_	170816EP02-19	
OC-42	Pared	7	_	✓	_	_	170810EP01-07	
OC-43	Pared	12	✓	✓	MN318308	XXXXXXX	170810EP01-08	
OC-44	Pared	9	_	✓	_	_	170810EP01-09	
OC-46	Pared	13	_	✓	_	_	170810EP01-11	
Muricea austera	OC-10	Acuario	7	_	✓	_	_	170816EP02-15	
OC-11	Acuario	12	_	✓	_	_	170816EP02-16	
OC-64	Cuarenta	9	✓	✓	_	XXXXXXX	170907EP07-09	
OC-35	Cuarenta	13	_	✓	_	_	170816EP07-14	
OC-36	Cuarenta	7	_	✓	_	_	170816EP07-15	
OC-59	Cuarenta	5	_	✓	_	XXXXXXX	170907EP07-04	
OC-9	Cuarenta	15	_	✓	_	_	170816EP02-14	
Heterogorgia hickmani	OC-18	Acuario	20	✓	✓	_	_	170816EP02-23	
OC-26	Cuarenta	12	_	✓	_	_	170816EP07-05	
OC-32	Cuarenta	10	_	✓	_	_	170816EP07-11	
OC-58	Cuarenta	18	_	✓	_	_	170907EP07-03	
OC-70	Cuarenta	15	✓	✓	MN318313	XXXXXXX	170907EP07-15	
Psammogorgia arbuscula	OC-37	Pared	10	_	✓	_	_	170810EP01-02	
OC-38	Pared	17	_	✓	_	_	170810EP01-03	
OC-39	Pared	12	_	✓	_	_	170810EP01-04	
OC-40	Pared	18	_	✓	_	_	170810EP01-05	
OC-41	Pared	14	_	✓	_	_	170810EP01-06	
OC-49	Pared	20	_	✓	_	_	170810EP01-01	
Leptogorgia alba	OC-55	Pared	8	✓	✓	MN318312	XXXXXXX	170810EP01-19B	
OC-65	Cuarenta	15	_	✓	_	_	170907EP07-10	
OC-54	Pared	12	_	✓	_	_	170810EP01-19A	
OC-66	Cuarenta	22	_	✓	_	_	170907EP07-11	
Lptogorgia cf. alba	OC-67	Cuarenta	18		✓	MN318311	XXXXXXX	170907EP07-12	
Leptogorgia obscura	OC-72	Acuario	12	_	✓	_	_	170816EP02-20	
OC-74	Acuario	10	✓	✓	MN318314	XXXXXXX	170816EP02-16	
OC-75	Acuario	15	_	✓	_	_	170816EP02-20	
Pacifigorgia rubicunda	OC-20	Acuario	12	✓	✓	XXXXXXX	XXXXXXX	170816EP02-25	
OC-21	Acuario	20	_	✓	_	_	170816EP02-26	
OC-68	Cuarenta	16	_	✓	_	_	170907EP07-13	

Morphological examination

Morphological features were examined using dry and ethanol subsamples, in situ images, and observations. Octocoral specimens were treated following the methodology described by Breedy & Guzman (2003b). For sclerite characterization, an Eclipse Ci-L electronic microscope equipped with a Nikon digital camera DS-Fi3 was used for the identification. The size of the sclerites was determined using an optical micrometer and measured using the software NIS-Elements D version 3.10. Sclerite characterization included dominant sclerite type, coenenchyme sclerite color, and sclerite measurements (length/width). External characterization was carried out on a dry sample of the whole colony and in situ images. The following characters were selected for their relevance from a taxonomic perspective: colony color, polyp color, colony shape and size (length/width), branching pattern, length of unbranched terminal branchlets, and diameter of end branchlets.

DNA extraction and sequencing

DNA of the octocoral species was obtained from 95% ethanol-preserved subsamples. DNA was extracted following the guanidine extraction protocol as described by Sinniger, Reimer & Pawlowski (2010). Two mitochondrial regions were targeted for PCR. Partial cytochrome oxidase subunit I (mt-COI) was amplified using the primers COII8068x-F and COIOCT-R (McFadden et al., 2011), while the variable mt-MutS region was amplified using the primers ND42599-F (France & Hoover, 2002) and MutS3458R (Sánchez et al., 2003). The raw data for the DNA sequences obtained in this study are available at https://www.ncbi.nlm.nih.gov/popset/1824127159?report=genbank.

PCR amplification conditions for both markers were as previously described by Sánchez et al. (2003), using standard Taq polymerase (Bioline). All PCR-amplified DNA fragments were observed on a 1.5% agarose gel stained with SYBR Safe using a UV light source. The PCR product was purified using the GeneJET PCR Purification Kit (Thermo Fisher Scientific) and submitted for sequencing in both directions (LGC Biosearch Technologies, Germany). It was not possible to amplify and sequence all the gene regions from all specimens. For instance, the mt-COI did not amplify the M. plantaginea, M. squarrosa, M. austera and P. rubicunda, while for the mt-MutS only M. plantaginea and M. squarrosa were not amplified (Table 1). However, for those sequenced, the resulting electropherograms were checked for quality and inconsistencies, and assembled using Geneious Prime 2019.1.1 (Kearse et al., 2012).

The BLAST algorithm search tool was obtained from GenBank (https://blast.ncbi.nlm.nih.gov/Blast.cgi) (Mahnir, Kozlovskaya & Kalinovsky, 1992) was applied to examine the consensus sequences to rule out contamination and to find similarities with other octocoral sequences. The sequences of the octocorals obtained in this study were uploaded into GenBank, and the accession numbers are listed in Table 1. Additional octocoral sequences for both loci were downloaded from GenBank and used for phylogenetic analysis.

Alignments

Geneious software was used to create two master alignments for both loci, with several octocoral sequences reported for the Eastern Pacific (Soler-Hurtado, López-González & Machordom, 2017; Vargas et al., 2014). The mt-MutS dataset included 96 octocoral sequences (11 sequences from this study and 85 sequences from GenBank) while the mt-COI alignment included only 31 octocoral sequences due to the inclusion of a few sequences reported for this locus in the TEP region (nine sequences from the present study and 22 sequences from GenBank). The complete mt-MutS alignment comprised 840 bp long and showed a higher number of nucleotide substitutions and indels than the full mt-COI alignment of 450 bp.

Phylogenetic analyses

Phylogenies were reconstructed to obtain a broader picture of the classification of the Ecuadorian octocorals. For the purposes of species classification, a set of species was selected which covered the three most important octocoral families, namely Gorgoniidae, Plexauridae, and Acanthogorgiidae. These were identified the course of the present study and as well as in the latest systematic studies of the TEP region (Soler-Hurtado, López-González & Machordom, 2017; Vargas et al., 2014). Additionally, two species from the monophyletic group of the genus Plexaurella were included as outgroups (Plexaurella grisea and Plexaurella dichotoma; accession numbers are included in both trees). A phylogenetic reconstruction was conducted using a maximum likelihood framework with RAxML 3.3.2 integrated into Geneious Prime 2019.1.1. For each dataset, a GTR (GTR + Г + I) model of nucleotide substitution (four rate categories) and a gamma model with 1,000 bootstrap interactions were employed. In this study, two maximum likelihood (ML) phylogenetic trees are presented, based on two mitochondrial datasets: the mt-MutS and mt-COI regions.

Metabolomic analyses

Sample preparation

A total of three to 12 replicates of each species were collected, depending on their abundance at the sites, from different depths and locations across the study area (see Table 1). The samples were subsequently frozen at –20 °C, freeze-dried, and grounded to generate a homogenous powder. For metabolomic analysis, samples were prepared according to Santacruz et al. (2019) with minor modifications. Briefly, a mass of 1 g of each sample was extracted three times with 10 mL of MeOH/CH2Cl2 (3:1) mixture for 4 min under ultrasound conditions. The extracts were filtered and combined, and 250 mg of C18 powder was added before evaporation to dryness using a SpeedVac instrument. Dried samples were recovered and loaded onto C18 Solid Phase Extraction (SPE) cartridges (2 g, Hypersep C18; Thermo Fisher Scientific). The SPE cartridges were cleaned with 10 mL of MeOH and then conditioned with 10 mL of H2O prior to sample loading. After loading the solid, the samples were first desalted with 10 mL of H2O and then eluted with 10 mL of MeOH/CH2Cl2 (3:1). The organic phase was evaporated to dryness using the SpeedVac. Each extract was redissolved in 1 mL of a mixture of solvents MeOH/CH2Cl2 (3:1). The resulting solutions were then filtered through a 0.2 μm filter and then placed in a 1.5 mL vial for UHPLC-HRMS analysis.

UHPLC-qToF analyses

Mass spectra (MS) fragmentation analyses were performed using a UHPLC (Agilent 1290) coupled to an Agilent 6540 UHD Accurate Mass quadrupole time-of-flight mass spectrometer (qToF). Purine C5H4N4 [M + H]+ (m/z 121.0509) and hexakis (1H,1H,3H-tetrafluoropropoxy)-phosphazene C18H18F24N3O6P3 [M + H]+ (m/z 922.0098) were used as internal lock masses. The MS fragmentation analysis was performed in positive ionization mode on an Acquity™ UPLC BEH Fluoro Phenyl Column, 130 Å, 1.7 μm, 2.1 mm × 100.0 mm (Waters), at a temperature of 40 °C. The mobile phase was composed of two solvents, designated A (H2O + 0.1% Formic Acid) and B (CH3CN + 0.1% Formic Acid). The injection volume was set at 3 μL, and the elution rate was 0.4 mL.min−1. The elution gradient was programmed in the following manner: The mobile phase consisted of 90% solvent A, 10% solvent B for 2 min, followed by an increase in B with a linear gradient up to 100% from to 2 to 10 min, 100% B for 2 min, returning to the initial condition from 12 to 13 min. Finally, 2 min of post-run for column equilibration, resulting in a a total run time of 15 min.

Untargeted metabolomics

A total of 71 samples were analyzed through untargeted metabolomics and they are listed in Table 1. The MS parameters were set as follows: nebulizer gas N2 at 35 psi, gas temperature: 300 °C, drying gas N2 at 8 L/min, ion source Dual AJS ESI; ToF spectra acquisition from 100 to 1,700 amu, scan rate 3.00 spectra/sec; and capillary voltage 3,500 V. A quality control (QC or pools) sample was prepared from an aliquot (20 μL) of each sample. QC samples were injected five times at the beginning of the batch, five times at the end of the batch, and once after each of the five samples. The sequence started with three injections of the methanol (blank) samples and three injections at the end immediately before each QC sample to detect any type of contamination in the column during the sequence. LC-MS raw data files were converted into mzML files using MSConvert in the ProteoWizard library (Kessner et al., 2008). All MS raw data is available through the repository Figshare (https://figshare.com/s/67d62277dc542c0f1d13).

For MS analysis, the mzML files were processed using the R package XCMS (R version 3.3.1., XCMS version 1.50.0) to identify, deconvolute, and align features (molecular entities with a unique m/z and a specific retention time) (Smith et al., 2006). The parameter settings for XCMS processing were set as described for UPLC-QTOF (high resolution) (Patti, Tautenhahn & Siuzdak, 2012). The generated matrix was filtered by removing peaks abundantly found in the blank samples (MeOH injections).

Multivariate data analyses

To assess the variability among genera, the octocoral specimens were organized into five different groups, each corresponding to the five genera (Muricea, Leptogorgia, Pacifigorgia, Psammogorgia, Heterogorgia). Additionally, to investigate the variability within the Muricea genus, a second matrix was organized into seven groups containing 51 specimens. Of these, 48 specimens belonged to six species of Muricea, while the remaining three specimens were assigned to the outgroup species, Leptogorgia obscura. The data were uploaded to the MetaboAnalyst 4.0 website (Chong et al., 2018) for the statistical analysis. To filter analytical dissimilarity and to explore the molecular ions with the highest intensities (major peaks), a quality criterion was established according to the peak intensities observed in the octocoral metabolomic profiles. Any variation of QC samples around their mean (CVQC) >20% was used for data analysis, as this was deemed to be an appropriate threshold for the purposes of this study. Missing values were replaced by the K-nearest neighbors (KNN) method, and no additional filtering was applied. The datasets were then median normalized, log transformed, and Pareto scaling was applied prior to statistical analysis, which was conducted using two multivariate analysis methods: the non-supervised principal component analyses (PCA), and the supervised partial least squares-discriminant analyses (PLS-DA). To assess the significance of class discrimination in the PLS-DA, a cross-validation test was initially performed at both the genus and species levels, whit the models were evaluated using R2 and Q2 metrics. The R2 value indicates the total proportion of total variance in the dataset that is explained by the model, including both the R2X and the independent R2Y variables. The Q2 value represents the model’s predictive capability, and the accuracy is measured by the ratio R2/Q2 (considered satisfactory when > 0.7) (Fig. S2 and Table S1).

To ensure the significance of the results, 100 permutations were performed. A dendrogram was obtained using hierarchical cluster analysis (HCA), in which two parameters were considered. The first parameter is the similarity measure, which comprises Euclidean distance, Pearson’s correlation, and Spearman’s rank correlation. The second parameter is the clustering algorithms, which include average, complete, and single linkages, as well as Ward’s linkage. Hierarchical clustering was performed with the htclust function in the stat package, which produce a dendrogram showing the distance measure using Euclidean, and the clustering algorithm using ward.D. These analyses were employed to investigate the similarities between the five different octocoral genera (71 specimens), and between the six octocoral species (48 specimens) of the genus Muricea.

Results

Morphological analyses

A total of 11 octocoral species were successfully identified based on their morphological features. The 12 specimens were identified as belonging to the three distinct families: Plexauridae, Acanthogorgiidae, and Gorgoniidae. For the family Plexauridae, the genus Muricea was found to be highly diverse and abundant, and six species were identified as M. crassa, M. fruticosa, M. plantaginea, M. squarrosa, M. austera and M. purpurea. In the family Acanthogorgiidae, only one species was identified for the genus Heterogorgia, H. hickmani. In the family Gorgoniidae, two species of the genus Leptogorgia were identified as L. alba and L. obscura. Furthermore, there is some uncertainty regarding the identity of OC-67 (Leptogorgia cf. alba), as it presented discrepancies in the color of the colony, the color and sizes of the coenenchyme and sclerites, and in the measurements of the colony and branches when compared to the original description of L. alba. However, morphological data is insufficient to confidently identify it as a new species. Therefore, it was tentatively named as Leptogorgia cf. alba. Finally, in the latter family, only one species was identified for the genus Pacifigorgia as P. rubicunda, and one species was identified for the genus Psammogorgia as P. arbuscula (Fig. 1). The morphological features of the 12 octocoral specimens are summarized in Table 2 and compared with literature data.

Table 2 Morphological features.

Comparative morphological features of the Eastern Pacific octocoral species found at REMAPE with similar species described in previous studies. Species in bold were identified in this study and measurements are given in cm/mm.

Species name (Sample code)	Colours of colony	Colours of the polyps	Colony shape	Colony size length/wide (cm)	Branching pattern	Length of
unbranched
terminal
branchlets (cm)	Diameter
of end
branchlets (mm)	Dominant sclerite type	Coenenchyme sclerites
colours	Sclerites size length/wide (mm)	
* Muricea fruticose
(OC-3)	Reddish to dark brown/white to pale yellow stems bicolored.	White	Bushy spherical	25 × 40	Irregularly	15–40	3–6	Unilateral spinose spindles	Deep reddish brown/
brownish
yellow to pale yellow	0.28–1.0 × 0.05–0.10	
M. fruticose
(Breedy & Guzman, 2016)	Reddish brown-white bicolored	White	Bushy	35 × 45	Irregularly	15–40	3–5	Unilateral spinose spindles	Reddish brown/ amber/pale yellow to whitish	0.30–1.0 × 0.05–0.12	
* Muricea plantaginea
(OC-53)	White-grey or light brown to yellowish	White or yellow	Branch in a single plane/ flabelliform	42 × 20	Irregularly lateral	10–50	2–3	Leaf like-Spindles	Reddish brown/ amber to white	0.25–1.0 × 0.08–0.21	
Muricea plantaginea (Breedy & Guzman, 2016)	Deep brown and white	Na	Flabelliform	40 × 18	Irregularly lateral	10–50	2-3	Leaf like-Spindles	Reddish brown and amber	0.22–1.0 × 0.09–0.20	
* Muricea austera
(OC-64)	Reddish brown/
pale yellow	Reddish orange to yellow	Bushy	17 × 20	Dichotomouslateral	50	6–9	Unilateral spinose spindles	Reddish brown/pale orange/ yellow	0.52–1.3 × 0.18–0.50	
Muricea austera (Breedy & Guzman, 2016)	Reddish brown	Na	Bushy	20 × 23	Dichotomous-lateral	50	7–8	Unilateral spinose spindles	Reddish brown/orange/light yellow	0.55–1.5 × 0.20–0.50	
* Muricea crassa
(OC-63)	Deep brown	White or yellow	Bushy	35 × 48	Dichotomous-lateral	70	7–10	Unilateral spinose spindles	Reddish brown/ lighter and darker	0.52–2.0 × 0.40–0.68	
Muricea crassa
(Breedy & Guzman, 2016)	Dark brown	Na	Bushy	40 × 50	Dichotomous- lateral	70	7–10	Large and irregular spindles	Reddish brown	0.56–2.5 × 0.40–0.70	
* Muricea purpurea
(OC-43)	Blue and purple	Reddish orange	Bushy	16 × 22	Dichotomous	50	12–14	Leaf like-Spindles	Dark red/reddish orange	0.3–0.60 × 0.09–0.28	
Muricea purpurea (Breedy & Guzman, 2016)	Reddish purple	Reddish orange	Bushy	22 × 21	Dichotomous	50–80	9–11	Leaf like-Spindles	Dark red/reddish orange	0.3–0.70 × 0.10–0.30	
* Muricea squarrosa
(OC-12)	Light to deep brown	Dull yellow to light brownish	Bushy/ flabellate	19 × 13	Dichotomous	40	12–18	Spindles	Pale yellow to light brownish	1.2 × 0.20	
Muricea squarrosa (Breedy & Guzman, 2015)	Light brown/brown	Na	Flabellate	14 × 12	Dichotomous	40	12–20	Spindles	Dull yellow to light brownish/ whitish and colourless	1.3 × 0.23	
* Leptogorgia alba
(OC-55)	Pale pink	White/colourless	Flabellate	12 × 11	Irregular/pinnate	3 or less	1–2	Spindles	Colourless and few pink	0.03 × 0.14	
*Leptogorgia cf. alba
(OC-67)	White pale	White/colourless	Flabellate	18 × 15	Irregular/pinnate	3 or less	1–1.5	Spindles	Colourless	0.05 × 0.18	
Leptogorgia alba, (Breedy & Guzman, 2007)	White	White/colourless	Flabellate	9.5 × 11	Dichotomous/ irregular	3 or less	1.0–1.5	Spindles	Colourless and few pink	0.04–0.06 × 0.18	
Leptogorgia manabiensis
(Soler-Hurtado et al., 2017)	Dark pink	White/colourless	Flabellate	15.5 × 12	Irregular/
pinnate	3 or less	1.0–1.9	Spindles, straight or bent	Colourless and few pink	0.06–0.17 × 0.03–0.06	
* Leptogorgia obscura
(OC-74)	Purple	Purple to red	Irregular/ pinnate	10 × 15	Dichotomous	50	1–3	Spindles/ capstans	Violet/pink	0.06–0.12 × 0.02–0.05	
Leptogorgia obscura (Breedy & Guzman, 2007)	Dark violet	Violet to pink	Irregular/ pinnate	4 × 4	Dichotomous	52–54	1.5–2	Capstans	Violet/pink	0.08–0.12 × 0.05	
* Heterogorgia hickmani
(OC-70)	Beige to yellowish or greenish	Bright yellow	Incrusting holdfast	15 × 12	Unbranched stems	8–9	10–12	Curved spindles	White to colourless	0.50–0.70 × 0.05–0.14	
Heterogorgia hickmani (Breedy & Guzman, 2005a)	Whitish to beige or greenish in ethanol or dry preserved	Bright yellow	Na	18 × NA	Unbranched stems	13	10–13	Curved spindles	White to colourless	0.50–0.66 × 0.05–0.13	
* Psammogorgia arbuscula
(OC-41)	Greenish to red orange	Pale pink/ yellowish	Bushy or flabellate	25 × 18	Dichotomous	1–1.5	3–6	Warty spindles	Coral red/some light orange	0.19 × 0.07	
Psammogorgia arbuscula
(Breedy & Guzman, 2014)	Red, orange, pink or white	Na	Bushy or flabellate	Na	Dichotomous-irregularl/ dichotomous or supinate	Na	Na	Warty spindles, radiates and crosses	Red orange	0.30 × 0.14	
* Pacifigorgia rubicunda
(OC-20)	Deep orange to yellowish	Pale orange to yellow	Thick and stiff	12 × 16	Several fans	Absent or very small	Short 2–3	Blunt spindles	Orange to yellow/ colourless or bicoloured	0.06-0.012 × 0.02–0.04	
Pacifigorgia rubicunda (Soler-Hurtado et al., 2016)	Brownish orange	Na	Fan	4 × 16	Several fans	Na	Short 0.8–1	Blunt spindles	Pink, orange and lemon yellow bicoloured and multi-coloured	0.1 × 0.04	
Notes:

Na = data not available.

* In bold specimens from this study.

Phylogenetic analyses

The mt-MutS variable region was successfully sequenced for 11 octocoral specimens attributed to nine different species, while the mt-COI was successfully sequenced for eight octocoral specimens assigned to eight different species (Table 1). The BLAST results from the 19 sequences matched with octocorals (families Plexauridae, Acanthogorgiidae and Gorgoniidae) reported for the TEP region. The phylogeny reconstructed using ML for mt-MutS is shown in Fig. 2, and the ML mt-COI tree is shown in Fig. 3A. For the families Plexauridae and Acanthogorgiidae, phylogenies drawn from both sets of data identified OC-70 as a member of Heterogorgia (Vargas et al., 2014). In the mt-MutS tree, our specimen was assigned closer to H. verrucosa with only two different base pairs and four base pairs to H. hickmani as reported by Vargas et al. (2010). A similar situation was observed in the mt-COI tree, with the difference that no COI sequence for H. hickmani was available for comparison. Both mitochondrial trees showed highly supported values (100% bootstrap support for MutS and 98.6% bootstrap support for COI) for Heterogorgia species.

Figure 2 Maximum Likehood Phylogenetic tree obtained from sequences of mitochondrial MutS DNA (mt-Muts-DNA).

LM bootstrap values over 70% are indicated by nodes. Values below 70% were considered as unresolved. Specimens from this study are indicated in bold.

Figure 3 Molecular and metabolomic comparative analyses.

(A) Maximum likehood phylogenetic tree obtained from sequences of mitochondrial cytochrome oxidase subunit 1 (mt-COI). ML bootstrap values over 70% are indicated by the nodes. Values below of 70% were considered as unresolved. Specimens from this study are indicated in bold and highlighted with different figures and colours. (B) Untargeted metabolomics for Muricea species by HCA analyses. Clustering result shown as dendrogram between the 48 metabolomic profiles of the Muricea species; M. austera, M. crassa, M. fruticosa, M. plantaginea, M. purpurea, M. squarrosa and three metabolomic profiles of Leptogorgia obscura as an outgroup. Specimens from this study are indicated in bold and highlighted with different figures and colours (distance measure using euclidean, and clustering algorithm using ward.D.).

The specimens identified as Muricea crassa (OC-63, OC-56), M. fruticosa (OC-3), M.austera (OC-64, OC-59), and M. purpurea (OC-43) clustered with other Muricea species recently reported in the TEP region (Figs. 2 and 3A) (Soler-Hurtado, López-González & Machordom, 2017; Vargas et al., 2010, 2014). Low resolution was observed for both loci (mt-MutS and mt-COI) for this genus, preventing a clear distinction between Muricea species. However, both markers supported the identification at the genus level, with high bootstrap values of 100% for the Muricea clade. Despite the low resolution of both loci, small groups were observed within the Muricea clade of the mt-MutS tree. The first group included the species M. crassa and M. fruticosa with a small difference (4 bp) between both species, and the second group contained the species M. austera and M. purpurea, and no difference was found between the sequences. The latter subclade reflects the morphological similarities between M. purpurea and M. austera (Table 2). Finally, a third group was formed with other Muricea species, in which only M. muricata was distinct. Furthermore, the mt-COI MLphy tree revealed that all Muricea sequences clustered together without any discrimination at the species level. Only one specimen of M. fruticosa exhibited a minor difference of six base pairs from the other species. These findings provide evidence to support the hypothesis that this locus has lower resolution than the mt-MutS region.

Within the family Gorgoniidae, the sequences obtained for both mt-MutS and mt-COI for Leptogorgia obscura (OC-74), L. alba (OC-55, pink morphotype) and L. cf. alba (OC-67, white morphotype) clustered with other species of the same genus. L. obscura (OC-74) was consistent with a previous L. obscura reported by Soler-Hurtado et al. (2016), and this small clade was supported on the COI tree (74.7%) (Fig. 3A). Data from both mitochondrial loci indicated that L. alba (OC-55) and L. cf. alba (OC-67) were distinct. The small clade containing L. alba (OC-55) in the MutS tree was strongly supported by bootstrap analysis, with a value of 88%. In contrast, L. cf. alba (OC-67) was placed in a distinct subclade with bootstrap values greater than 70%, indicating a high degree of confidence in its placement. Furthermore, the species reported as L. cofrini (Vargas et al., 2014) was placed in the same group as L. alba (OC-55) in both trees, and the sequences were identical. All Leptogorgia species from the TEP region form a monophyletic group with strong support (mt-MutS, 99%; mt-COI, 89.6%) (Soler-Hurtado, López-González & Machordom, 2017).

Finally, the mt-MutS sequence from Pacifigorgia rubicunda (OC-20) clustered in a small clade with other Pacifigorgia species, which was highly supported by a bootstrap value of 99%. No significant differences were found between P. rubicunda sequences; only four base pairs differed between P. rubicunda (OC-20) and the single sequence reported for P. rubicunda by Soler-Hurtado, López-González & Machordom (2017). However, the sequence from OC-20 (P. rubicunda) contained several indels compared to other Pacifigorgia species in the TEP realm. Despite these minor sequence differences, the results revealed the low-resolution potential of the mt-MutS region for separating species within the genus Pacifigorgia.

Despite several attempts, we could not amplify the mt-COI gene of the species M. austera (OC-64 and OC-59) and P. rubicunda (OC-20), and neither locus for the species M. plantaginea (OC-50, OC-7), M. squarrosa (OC-15, OC-16), P. arbuscula (OC-38).

Metabolomics analyses

Untargeted metabolomics at the genus level

The quality control samples used in the non-targeted metabolomics approach showed very low variability compared to the variability between groups, thus enabling reliable statistical analysis of the results. With the main aim of assessing the significance of the separation between groups and to flag some variable importance in projection (VIP), we decided to perform supervised PLS-DA (Fig. 4A).

Figure 4 Untargeted metabolomics at the genus level.

(A) PLS-DA of the untargeted metabolomic analyses at the genus level for Heterogorgia, Leptogorgia, Muricea, Pacifigorgia, Psammogorgia. The variances are shown in brackets and ellipses show a 95% of confidence. (B) Highest variable importance in projection (VIP) scores at genus level. The column on the left part indicates the chemical feature name combined by two important features (M = Parent mass integer and T = Retention time in sec) identified by PLS-DA analyses. The coloured boxes on the right indicate the intensity level of the corresponding chemical feature in each octocoral genera under study; Heterogorgia, Leptogorgia, Muricea, Pacifigorgia, Psammogorgia.

A total of 3,595 molecular ions (m/z and rt) were obtained after performing a non-targeted metabolomic approach using UHPLC-HRMS on 71 octocoral samples (six replicates for each species), including 14 samples for morphological assessment and molecular studies. The metabolomic profiles exhibited consistency between all replicates within a species, while significantly differences were observed between the 12 species under study, and between the five genera (Figs. S4–S15). Supervised PLS-DA of the molecular ions distributed among the five groups showed a high level of similarity within a genus (Fig. 4A). However, using only the two main components, Muricea and Leptogorgia were not well separated in the PLS-DA analysis. Overall, the profiles of the five genera were significantly dissimilar, as demonstrated by the low permutation score (p < 0.01).

HCA was performed to investigate the similarity between the 71 metabolomic profiles from the five genera (Fig. S1). Each genus formed a separate cluster, with Muricea being the genus with the most distinct metabolome, followed by the Leptogorgia and Psammogorgia clusters, which were more similar, and finally, Heterogorgia and Pacificgorgia. All specimens belonging to the same genus were more similar than two specimens belonging to two different genera, confirming the excellent permutation score of PLS-DA and the efficiency of the metabolomics approach in distinguishing genera (Fig. 4A). The cross-validation test of PLS-DA also revealed high scores for R2 and Q2 with an accuracy higher than 0.7 considering only the principal component (Fig. S2). The three values were clearly significant using the three principal components of the PLS-DA.

To identify some metabolites responsible for the separation between the different genera, we focused on the 15 molecular ions with the highest scores (>3.2) among the VIPs (Fig. 4B). Among the 15 VIPs, the13 molecular ions exhibiting the highest intensity (in red) were detected in the Heterogorgia (5) and Psammogorgia (8) samples. Unfortunately, the lack of chemical studies on both genera have prevented accurate identification of these molecular ions. However, two of the high-intensity molecular ions detected in Heterogorgia, as well as in Leptogorgia species (M388T165 and M344T129), were putatively identified as furanocembranoids by comparison with previously reported compounds from Leptogorgia setacea (Ksebati, Ciereszko & Schmitz, 1984) and Leptogorgia sp. (Gallardo et al., 2018). Furthermore, the mass spectra (HR-MS1) of octocoral samples were manually inspected to tentatively identify muricin 4 from M. fruticosa, leptodienone B from L. alba and leptolide from L. cf. alba, Table 3 following the general accepted guidance described by Alseekh et al. (2021). The MS data of these compounds were then compared with those reported in the literature (Bandurraga & Fenical, 1985; Ortega et al., 2008; Gutiérrez et al., 2005), as well as in databases such as Chemspider and the metabolomic work bench database.

Table 3 Molecular ions identified from octocoral species.

Summary of the molecular ions identified from the genera Muricea, Leptogorgia.

Organism	Rt (min)	Putative metabolite	Molecular formula	Ion Type	ES (+) Measured m/z	ES (+) Theoretical m/z	m/z error (ppm)	Reference	
Muricea fruticosa	10.57	muricin 4	C39H59NO9	[M + H]+	686.4254	686.4190	9.6151	Bandurraga & Fenical (1985)	
Leptogorgia alba	8.58	leptodienone B	C20H30O2	[M + H]+	303.2266	303.2245	6.9255	Ortega et al. (2008)	
Leptogorgia cf. alba	6.28	leptolide	C20H22O6	[M + H]+	359.1439	359.1416	6.4041	Gutiérrez et al. (2005)	

Additionally, we aimed to assess the potential of the metabolomics approach to classify the genera and to compare this metabolomic classification with a phylogenetic tree. Overall, the classification of the genera according to metabolomic data did not agree with the phylogenetic relationships proposed in this study (Figs. 2, 3A). For example, in metabolomic HCA, the genus Leptogorgia was closely related Psammogorgia rather than Pacifigorgia (Fig. S1) while, on both phylogenies (mt-COI and mt-MutS) presented above, Leptogorgia appeared as a sister clade to the genus Pacifigorgia. Likewise, specimens from the genus Muricea were very distinct from those four genera via metabolomics, whereas phylogenetic analyses using molecular barcoding sequence demonstrated that they should be closer to Heterogorgia. However, differences in metabolomic profiles among the three species from the genus Leptogorgia; L. alba (OC-55, 65, 66), L. cf. alba (OC-67), and L. obscura (OC-74, 75, 72) were evident (Fig. S1). This outcome was supported by both phylogenies (mt-COI and mt-MutS). Combining these data with morphological analysis, we suggest that L. cf. alba (OC-67) and L. alba are distinct species.

Untargeted metabolomics for Muricea species

A second metabolomic analysis was conducted to separate Muricea species to evaluate the potential of untargeted metabolomics at the species level. Therefore, only 51 samples (with a minimum of three and a maximum of 12 replicates for each species) were used; 48 specimens belonging to the six species of the genus Muricea (M. fruticosa, M. squarrosa, M. plantaginea, M. purpurea, M. crassa, M. austera) and three specimens of L. obscura were included as an outgroup. The PLS-DA analysis did not provide a supported separation between the six Muricea species (Fig. S3). However, the HCA revealed that the chemical profiles of all replicates within a species were similar, while significant dissimilarities were observed between the six Muricea species and the outgroup species L. obscura (p < 0.01) (Fig. 3B). In this case, the accuracy was lower than 0.7, indicating a lower predictability of the model. Consequently, the VIPs scores for the Muricea species were considered not representative, as these molecular ions are assumed to represent minor metabolites. Therefore, no chemical markers were identified in this study for the purpose of species-level separation.

Although muricin 4 (steroidal glycoside) was putatively identified from M. fruticosa (Table 3), the presence of fragmentation patterns resulting from the loss of H2O (M+-18), which is characteristic of hydroxylated sterols and steroidal glycosides, suggests that these compounds are present in Muricea species. It has been previously reported that certain sterols, including 19-nor-cholest-4-en-3-one, dinosterol, and muriceanol, as well as several saponins, are present in Muricea species (Popov, Carlson & Djerassi, 1983; Bandurraga & Fenical, 1985; Murillo-Álvarez & Encarnación-Dimayuga, 2003; Lorenzo et al., 2006; Gutiérrez et al., 2006).

Different conclusions can be drawn regarding the potential of the metabolomics approach to classification. The chemical profiles of M. austera, M. crassa, M. fruticosa and M. purpurea were closely related and distant from those of the two other and well-separated species, M. squarrosa and M. plantaginea. It is possible that some molecular ions detected in the latter two Muricea species may have acted as PCR inhibitors, thereby preventing an accurate separation of the species. The presence of PCR inhibitors in octocoral species, including polysaccharides, polyphenols, and other metabolites, has been documented (Wong et al., 2024). This suggests the need for alternative protocols to improve DNA extraction. This provides an explanation for the difficulties encountered during the gene amplification in both species (M. squarrosa and M. plantaginea). The molecular ions detected in other Muricea species appeared to be shared among all species, and thus did not appear to be specific biomarkers. When comparing the chemical feature dendrogram (HCA) with the phylogenetic tree built in Figs. 3A–3B some similarities were observed. For example, M. austera and M. purpurea were identified within the same clade through the DNA barcoding and phylogenetic analysis, and they also exhibited a close relationship with the chemical HCA. This result illustrates the potential of metabolomic analysis to distinguish closely related sister species, which have not been previously revealed through phylogenetic analysis.

Discussion

In the current study, we assessed the diversity of octocoral group Malacalcyonacea, representative invertebrates in the shallow waters of the REMAPE area, using an integrative taxonomy approach. Current taxonomic approaches applied to 71 octocoral specimens allowed us to identify 11 known species and one potentially new species currently identified as Leptogorgia cf. alba. The identified octocorals belong to the three families Gorgoniidae (L. alba, L. obscura, P. arbuscula and Pacifigorgia rubicunda), Plexauridae (M. plantaginea, M. squarrosa, M. purpurea, M. austera, M. crassa, M. fruticosa) and Acanthogorgiidae (H. hickmani). These species have already been reported in nearby areas of this region, such as Panama, Colombia, Costa Rica, and in other locations along the Ecuadorian coast (Abad et al., 2022; Breedy & Guzman, 2005a, 2003b; Breedy, Guzman & Vargas, 2009; Glynn, 2003; Soler-Hurtado, López-González & Machordom, 2017; Williams & Breedy, 2004). However, for the first time, metabolomic data were added to strengthen the identification of the species and improve species delimitation in cases where molecular analysis alone was unable to discriminate. Furthermore, this study contributes new records of these species to our region, including new molecular and chemical data.

The phylogenetic relationships among the octocoral species presented here were consistent with those reported in previous phylogenetic studies in the TEP realm (Soler-Hurtado, López-González & Machordom, 2017; Vargas et al., 2010, 2014). Both the mitochondrial regions were successfully amplified in most octocoral species. The higher resolution of mt-MutS over the mt-COI locus makes it a more suitable marker for identifying octocoral species (order Malacalcyonacea), as demonstrated previously (Soler-Hurtado, López-González & Machordom, 2017; Vargas et al., 2014). The molecular phylogenetic analyses presented here revealed that many inconsistencies remain in the separation of octocorals at the species level, similar to the species in the genus Muricea. This is a consequence of both the low resolution of the mitochondrial markers reported by Vargas et al. (2014) and the lack of accessible sequences in the databases. However, we employed these markers because our objective was to identify and confirm the identity of Ecuadorian specimens. To achieve a more accurate separation of sister species, we decided to combine phenotypic characteristics, such as morphological features and metabolomics, with molecular barcoding sequences. The metabolomics approach proved to be an efficient method for the separation of all genera, as well as the different species of Muricea, demonstrating its value as an additional dataset for species characterization. However, in terms of their suitability for classification, the results are controversial. An untargeted approach did not yield a good match with the phylogenetic trees of the targeted genera. However, the high level of similarities between sister species, such as M. austera and M. purpurea, and L. alba and L. cf. alba, was consistent with morphological data and molecular barcoding sequence.

By applying this integrative systematic methodology, we successfully identified a potentially new species within the Leptogorgia genus. The case of L. cf. alba provides an excellent illustration of the value of an integrative taxonomic approach. Indeed, the morphology did not provide clear separation of the L. alba specimens. Consequently, they were first described as distinct morphotypes based on variations, such as the color of the colony and sclerite sizes. Molecular analyses placed them in the same clade; however, the sequences of L. cf. alba (OC-67) and L. alba (OC-55) differed. The distinct species status was supported by metabolomic analyses, which also demonstrated a clear separation. Although only two compounds were putatively identified in both L. alba and L. cf. alba species (leptodione B and leptolide, respectively), the presence of unknown molecular ions indicates the need for further chemical studies to properly characterize their chemical composition in both species. Dorta et al. (2007) suggest that furanocembranolides could be used as chemotaxonomic markers for octocorals, depending on the degree of oxidation at C-18. In this context, we propose that different types of diterpenes can be considered as biomarkers of octocorals. Based on the aforementioned results, it was concluded that 12 species were correctly identified in this study, including one potential new species (L. cf. alba) described in detail herewith. We believe that a combination of molecular and metabolomic analyses has the potential to reveal new species within closely related morphotypes.

The metabolomic results for the four Muricea species, including M. crassa, M. fruticosa, M. purpurea and M. austera, revealed a degree of consistent with the molecular and morphological data. M. crassa and M. fruticosa were placed close to each other, and a similar situation was observed between M. purpurea and M. austera (see Figs. 3A, 3B and compare with Fig. 2). In the latter case, the metabolomic data had a higher resolution than the molecular barcoding sequence, and combined with distinct morphological features, we can infer the presence of the two species M. austera and M. purpurea. This case illustrates how the combination of morphological and metabolomic data enables the distinction of sister species that are difficult to separate using the existing molecular barcoding sequence method.

In terms of classification, one of the main problems raised by metabolomic studies is the lack of the chemical marker identification, which makes it very difficult to assess the potential for systematics. For further analysis, we propose the inclusion of molecular network chemometrics to circumvent this key issue. Indeed, it could allow the grouping of families of natural products and mapping of a network by taxonomic groups to quickly visualize the synapomorphic characteristics of the groups. A trait shared by two or more taxonomic groups has been derived through evolution from a common ancestral form. In this case, the specialized metabolites are considered to be the final products of the expression of unique biosynthetic gene clusters (Reverter et al., 2018). This study identified several putative molecular ions from Muricea and Leptogorgia species. The presence of unreported molecular masses, particularly from Muricea species, encourages further investigation of the chemical composition, metabolite characterization, and biological properties of the octocoral samples analysed.

The development of new tools in natural product chemistry, such as untargeted metabolomics using MS data, enables quick assessment of the broad chemical content of an organism owing to its high sensitivity and short working time. Taxonomy is known to be highly challenging in some complex groups, such as Cnidarians and Porifera, and metabolomic methods have already found some applications as an integrative methodologies for the taxonomy of those groups (Aratake et al., 2012; Cachet et al., 2015; Jaramillo et al., 2018; Santacruz et al., 2019; Tribalat et al., 2016; Vohsen, Fisher & Baums, 2019; Reveillaud et al., 2012). For example, an HPLC-MS-targeted metabolomics tool revealed the presence of different chemotypes (cembranoid diterpenes) associated with various soft corals of the genus Sarchophyton (Aratake et al., 2012). Jaramillo et al. (2018) demonstrated that the use of both an untargeted and targeted metabolomics methods with UHPLC-HRMS was an efficient approach not only as a complementary tool for the taxonomy of Zoantharians, as well as for their classification. Indeed, certain specialized metabolites, such as zoanthamines or 2-aminoimidazole alkaloids, have been identified as key biomarkers for specific zoantharian species (Guillen et al., 2019), as demonstrated by other group (Vohsen, Fisher & Baums, 2019). Despite the extensive and diverse applications of untargeted metabolomics, annotation of chemical markers detected by mass spectrometry remains a significant challenge. In some cases, minor metabolites of limited taxonomic relevance may contribute to the separation of the groups (Quinn et al., 2016).

Recently, metabolomic studies have revealed changes in the metabolomic content of marine invertebrates in response to environmental factors, including geographical location and seasonal changes. For instance, Reverter et al. (2018) employed an untargeted metabolomics approach to investigate the temporal and spatial intraspecific variation in two Mediterranean Haliclona sponges. However, intraspecific variation in the metabolome was found to be relatively minor compared to interspecific variation. Another investigation was performed on two zoantharian species from the genus Palythoa along the Brazilian coast, which assessed geographical patterns vs. species-specific metabolomes (Costa-Lotufo et al., 2018). The combined results of the metabolome variability and the molecular network analyses demonstrated that the differences between intraspecific and interspecific metabolome variability were greater than previously assumed. This finding calls into question the use of this approach for the separation of Zoantharian species. It also highlights the need for studies on larger temporal and geographical scales when environmental changes become more significant.

Conclusions

The current study illustrates the potential of untargeted metabolomics for the separation of different octocoral genera from the order Malacalcyonacea at the REMAPE. Furthermore, the method has been shown to be effective in distinguishing between sister species, a finding that was corroborated by an independent dataset. This has been evidenced by the morphological features of the two species of Muricea, as well as the molecular analysis of Leptogorgia cf. alba, revealing the latter to be a new species. The chemical data proved to be a valuable resource for the characterization of species. The detected unknown molecular ions in octocoral species such as Muricea and Leptogorgia suggests that they may represent untapped sources of novel chemical scaffolds, which could be the subject of further investigation.

Supplemental Information

Supplemental Information 1 Supplemental Information.

Supplemental Information 2 MutS gene DNA sequences.

Supplemental Information 3 Mitochondrial COII-COI gene DNA sequences from Ecuadorian Octocorals.

We would like to express our gratitude to Prof. J. A. Sánchez of the Universidad Nacional de Los Andes in Bogotá, Colombia, for his assistance with the taxonomic confirmation of the Ecuadorian octocoral species. We would like to express our gratitude to Dr. G. Genta-Jouve for his valuable comments and suggestions, which have significantly enhanced this work. The assistance of C. Domínguez, MSc, is acknowledged for his contribution to the sample collection process, and Guillermo Reyes, MSc, is acknowledged for his help in the annotation process of the octocoral sequences. The authors extend their gratitude to Dr. M. M. Gordo, D. Rodriguez, and Prof. O. P. Thomas of the Marine Biodiscovery laboratory at the School of Chemistry, National University of Ireland, Galway, for their valuable support in the metabolite extraction procedure.

Additional Information and Declarations

Competing Interests

The authors declare that they have no competing interests.

Author Contributions

Karla B. Jaramillo conceived and designed the experiments, performed the experiments, analyzed the data, prepared figures and/or tables, authored or reviewed drafts of the article, and approved the final draft.

Paúl O. Guillén performed the experiments, analyzed the data, prepared figures and/or tables, authored or reviewed drafts of the article, and approved the final draft.

Rubén Abad performed the experiments, analyzed the data, prepared figures and/or tables, authored or reviewed drafts of the article, and approved the final draft.

Jenny Antonia Rodríguez León conceived and designed the experiments, analyzed the data, authored or reviewed drafts of the article, and approved the final draft.

Grace McCormack conceived and designed the experiments, analyzed the data, authored or reviewed drafts of the article, and approved the final draft.

Ethics

The following information was supplied relating to ethical approvals (i.e., approving body and any reference numbers):

The University ESPOL under the Research Dean’s Department granted Ethical approval to carry out the study within its facilities (Ethical Application Ref: CENAIM-470-3-2014 and CENAIM-400-2019).

Field Study Permissions

The following information was supplied relating to field study approvals (i.e., approving body and any reference numbers):

Field experiments and permits were approved by the Ministry of Environment, Water and Ecological Transition of Ecuador (MAATE). (Project number: N°005-17 IC-FAU-DPSE/Maas).

DNA Deposition

The following information was supplied regarding the deposition of DNA sequences:

Octocorals sequences are available at GenBank: MN318307 to MN318314.

Data Availability

The following information was supplied regarding data availability:

All raw data from metabolomics analyses are available at Fisghare: Jaramillo, Karla B.; Guillen, Paúl O. (2025). MS-Untargeted Metabolomics for Octocorals-ECUADOR. figshare. Dataset. https://doi.org/10.6084/m9.figshare.25153277.v2.

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
