# Peer review of "Contribution of metabolomics to the taxonomy and systematics of octocorals from the Tropical Eastern Pacific"

_PeerJ, doi:10.7717/peerj.19009_

## Round 0.1 · original submission · Major Revisions

First, let me apologize for the time that it has taken to find expert reviewers to evaluatey your manuscript. Below and in an attached PDF you will find the detailed comments and suggestions from two reviewers Please ensure that you respond to each comment in a letter that clearly shows the changes made citing the corresponding line numbers. If changes are not made they need to be justified. In particular, please note the comment by reviewer 2 about this being presented as a case study. Perhaps you can find a way to frame this research in a different way as the data are very interesting and deserving of publication.

·

Basic reporting

In the present manuscript, the authors applied an integrative taxonomy method to identify and classify 12 Octocoral species (5 genera) of the REMAPE (Marine Protected Area El Pelado) from the Tropical Eastern Pacific near Ecuador. For that purpose, morphologic and DNA barcoding analyses were implemented along with metabolomic analyses on a total of 71 collected samples. The objective was to assess the potential of comparative metabolomics combined with multivariate analyses to support complementarily the discrimination and identification of different Alcyonacean samples down to the species level. Metabolomic analyses were shown to contribute to species discrimination with identification of a new species when other analyses such as genomic and morphologic analyses provided unclear results. The benefit of such integrated taxonomy method was illustrated in the manuscript with Muricea species and with Leptogorgia cf. alba, respectively. This is an interesting manuscript showcasing a taxonomic toolbox for the description and identification of such keystone marine species. The transparency and rigorous raw data sharing will increase biological and chemical knowledge on this Octocoral species.

The article is written in correct English, nevertheless a few sentences will require clarification and will need to be reformulated (see additional comments).

Introduction:
A few references are missing pertaining to the integrative taxonomy and chemotaxonomy of Octocorals (as suggested below).

Line 84 : The author could give the definition of what integrative taxonomy is (as opposed to chemo-taxonomy). Such explanation would also give more strength on the originality of the current study with the use of metabolomics.

Suggested additional references:
• Gerhart, Donald J. “The Chemical Systematics of Colonial Marine Animals: An Estimated Phylogeny for the Order Gorgonacea Based on Terpenoid Characters.” The Biological Bulletin 164, no. 1 (February 1983): 71–81.

• Coll, John C. “The Chemistry and Chemical Ecology of Octocorals (Coelenterata, Anthozoa, Octocorallia).” Chemical Reviews 92, no. 4 (June 1992): 613–31.

• Imbs, A. B., and T. N. Dautova. “Use of Lipids for Chemotaxonomy of Octocorals (Cnidaria: Alcyonaria).” Russian Journal of Marine Biology 34, no. 3 (May 1, 2008): 174–78.

• Kessel, Gustav M, Philip Alderslade, Jaret P Bilewitch, Kareen E Schnabel, and Jonathan P A Gardner. “The Use of Integrative Taxonomy in Octocorallia (Cnidaria: Anthozoa): A Literature Survey.” Zoological Journal of the Linnean Society 198, no. 2 (May 25, 2023): 677–90.

The figures have to be improved. Fig 3 and 4 could be combined. A figure illustrating the complementarity of metabolomics with other taxonomic methods would bring more depth to the presented results. (see suggestions in the section ''validity of the findings'').

In general, the interpretation of metabolomics results through multivariate data analyses lacks precisions. In a few instances inappropriate vocabulary was used (see validity of the findings). The complete review is also herein attached.

Experimental design

The objective of the study is clearly stated in the introduction (lines 82-83, and lines 109-111). The objective is to assess the potential of metabolomics to complementarily discriminate Alcyonacean taxa down to the species level for Muricea using an integrative taxonomy approach.

The experimental design in terms of sampling strategy has been done rigorously notably with regards to biological replicates. Methods pertaining to sample extraction and metabolomic data acquisitions are adequately described.

• Line 229 a new subsection should be added with the title ‘’Multivariate data analyses’’.
• Lines 248-250. Methods and algorithm used to perform HCA should be added.

Raw data pertaining to the LC-MS based metabolomics profiling are freely provided here: https://figshare.com/s/67d62277dc542c0f1d13. The raw data accession is working properly. The Genbank accession codes in table 1 are also adequately assigned.

The table 2 compiling sample codes and their description allow researchers to trace back the available raw data down to their original species. This was done rigorously and is very important in the context of raw data sharing and research transparency.

To the best of my knowledge, this research article is within the Aims and Scope of the journal.

Validity of the findings

Addressed knowledge gap: The combination of metabolomic and chemometric analyses were shown to contribute to species discrimination and identification of a new species when other analyses such as genomic and morphologic analyses provided unclear results. The benefit of such integrated taxonomy method was illustrated in the manuscript with Muricea species and the Leptogorgia cf. alba, respectively. The authors also counterbalanced their findings by acknowledging that metabolomics analyses combined with multivariate analyses did not provide perfect adequacy with Phylogenetic tree obtained though appropriate DNA sequencing.

The figures illustrating the collected results should better convey the complementarity of metabolomics to other taxonomic analyses.

A panel figure illustrating side by side the results from one type of analyses as opposed to those obtained by chemometrics would help the reader in understanding such complementarity. This could be done with regards to the Muricea species: a phylogenetic tree could be presented in one panel and on the other the HCA or a heatmap obtained with MS-based metabolomic data. The same color code should be used for the different figures of the manuscript.

- Fig 3 and Fig 4 could be combined in one single figure panel. The PLSDA validation results of fig 3b could be placed in supporting information instead, also the validation results do not show the R2/Q2 values as specified in the material and method section. Table summarizing the cross-validation results would be better than the exported graph from Metaboanalyst web-page.

- The VIP table in fig 4 does not bring enough information to be presented alone, unless some feature annotations are provided. There is no explanation on what the feature names M..T means in the figure legend. The scores are obtained for feature projections on a PC axis whose identity has not been specified.

- On fig 5, the different sample names on the HCA are a too small to be read correctly.

In general, the interpretation of metabolomics results through multivariate data analyses lacks precisions. There is a misuse of the term metabolites or molecules in the article. What is presented throughout the manuscript correspond to chemical features, i.e signals detected by mass spectrometry that are molecular ions, adducts or in-source fragment ions of metabolites. One metabolite can provide different MS features, and since there is no annotation of the MS data, the authors can only use the term chemical feature.

Pages 14/15 with PLSDA analysis and VIP scores.
- Line 335 and line 373: A pairwise post-hoc permutational test would have been more accurate to evaluate the differentiation between each group: that is between species within the Muricea genus (fig4) and between samples of the Muricea and Leptogorgia genera (fig3).

- Line 342: Which PC axis was used for the projection? This was not indicated in the text nor in the figure legend.

- Lines 348 and 379: “Among the 15 VIPs, XY were highly expressed”..a feature is not expressed ( it is not a gene), a feature is detected with a level of intensity, you can say line 348 “Among the 15 VIPs, 13 detected features were proportionally the most intense’’.

- Lines 350-351, Lines 380-381: an annotation of features (as these are not compounds at this stage) could have been provided instead. Annotation requires at least to provide a table with the measured and calculated m/z, the retention time the types of ion detected (e.g. [M+H]+, M+NH4]+) and the proposed molecular formula of the detected ion. Such table can already bring some hints as per the possible identity of the detected ion and thus possibly molecule.

- Lines 389-390: ‘’the VIP analysis showed that both species M. plantaginea and M. squarrosa produced distinct metabolites from other Muricea species” again here at this stage, the VIP does not show any differences in terms of metabolites presence of absence but only in terms of chemical feature relative intensities…. So the authors can only say ‘’ The VIP analysis shows that the composition in terms of chemical features is different between M. plantaginea and M. squarrosa’’

- Lines 391-392: ‘’These very specific metabolites could explain the difficulties encountered during the gene amplification of both species.’’ The authors can’t say this as they don’t know what these possible metabolites are. They should explain why they think possible metabolites may have affected the PCR process.

Additional comments

• The abstract mentioned “Additionally, this approach allowed a quick selection of putative bioactive compounds for further MS - guided isolation work among the metabolome of species of interest” This has not been illustrated in the results section of the manuscript. It seems to be part of an additional objective of the study that has not been performed here in the submitted manuscript.

• Introduction, Line 109: The authors mentioned ‘’ the first objective…’’ but in fact there is no ‘’second objective’’ so the word first could be removed.

•Discussion: The authors could also discuss their results with regards to the literature pertaining to chemotaxonomical investigations of Octocorals and thus propose some hypothesis as per the identity of discriminant features is concerned. What type of molecules are expected to be possibly discriminant in the analyzed Octocoral samples? The authors could mention again that previous results have shown that lipids, steroids and terpenoids were identified as discriminant metabolites for other Octocoral species (see also remarks pertaining to the literature).

•Discussion, Page 17 line 458: The authors could briefly define what a synapomorphic character is in order to better understand why molecular networking methods would help in identifying such synapormorphic metabolites.

•In the supporting information: A series of figures S8-154 were added at the end of the document but are not presented or mentioned in the manuscript, and, thus, do not have their place in the context of this manuscript.

•In general, the article is written in correct English. Nevertheless, a few sentences will require to be reformulated for more clarity as noted below:

Page 12 Line 264: “Therefore, it has tentatively named as Leptogorgia cf.alba’’ please correct as: ‘’Therefore, it has been named tentatively as Leptogorgia cf. alba’’

Page 15 lines 362: This sentence is strange in my opinion: “On the other hand, the dissimilarities between the metabolomic profiles of the three species from the genus Leptogorgia (OC-55, 65, 66, L. alba and OC-67, L. cf. alba and OC-74, 75, 72 L. obscura) were separated, and this result was well supported on both phylogenies (mt-COI and mt-MutS).’’ I would suggest the following “On the other hand, the differences in the metabolomic profiles among the three species from the genus Leptogorgia (OC-55, 65, 66, L. alba, OC-67, L. cf. alba, and OC-74, 75, 72 L. obscura) were evident, and this outcome was well supported by both phylogenies (mt-COI and mt-MutS).”

Page15 lines 396-398: “This result highlights a potential of the metabolomic analysis to distinguish, but place closely in the HCA, sister species that the molecular analysis could not reveal so far.”
The sentence is not clear and could be reformulated. For example, ‘’This result highlights the potential of metabolomic analysis to distinguish closely related sister species, which molecular analysis has not been able to reveal thus far.’’

Page 16 lines 413: “….species delimitation where molecular data could not discriminate”. Please modify for more clarity: “…species delimitation in cases where molecular data alone could not discriminate”

Page 17, Lines 468-473: Please reformulate this sentence is too long and something is missing: “In Jaramillo et al (2018), using both a untargeted and targeted metabolomic approach with UHPLC-HRMS was demonstrated that this method was very efficient as a complementary tool in zoantharian taxonomy but also for their classification and some specialized metabolites like zoanthamines or 2-aminoimidazole families of natural products identified as key biomarkers for certain species as shown for other groups by Vohsen,Fisher & Baums (2019).”

Reviewer 2 ·

Basic reporting

The manuscript would benefit from a thorough editing.
Specifically:
I would suggest to try to start off each paragraph with a short introductory sentence to make it easier for the reader to understand what the following part of the manuscript is about. I would refrain from starting paragraphs with words as “then” l.156 or “subsequently” l.162.
Please double check sentence structure and start of sentences. I would advise to refrain from starting sentences with “And” (e.g. line 342 or 345).
Use clear language when communicating results, I would refrain from using words as “some” to describe as in line 54 “ in some temperate and tropical waters” line 27 “ in some cases”
Make sure the focus of the sentence is clear:
e.g. line 50-51 “Beside stony corals, this group of marine invertebrates has therefore been the focus of interest of several studies” This sounds like the sentence focuses on stony corals, however the previous sentence focused on octocorals which are referred to as “this group” in the sentence presented. I would advise to make the focus on the sentence more clear e.g. “This group of marine invertebrates has therefore been the focus of interest of several studies, besides stony corals.”

The introduction and background puts the manuscript in context and the literature is well referenced. Structure of the manuscript conforms to PeerJ standards and the figures are relevant and clear.

Experimental design

This particular paper is partially within the scope of the journal. PeerJ considers Research Articles and systematic meta analyses, but the journal states that case studies are not an accepted format. The here presented manuscript is described as a case study.
The research question is well defined, relevant & meaningful, since integrating multiple variables and approached to answer research questions is a way forward. In this study the authors integrate metabolomics data together with genetics and morphological data to identify genera and species of the Alcyonacea order.
The investigation covered a good number of specimen across the well defined research area and the methods used were clear and of high standard. However there are a few questions left in terms of the specifics of the used methods that I would like to address:
-line 118: The samples were taken during May and September, how are potential temporal & spatial differences accounted for?
-line131-132: What are the minor modifications to the method?
-line 152-153: Please specify.
-line 184-209: Where does the sample preparation and measurement process come from? Was this method developed alongside with this study or is this a already published protocol? Why did you decide to extract with a solvent mixture of MeOH/CH2Cl2?
-line 253-237: Please add some more information why a threshold of 20% was used here.
-line: “as the number of features did not reach 5,000” please clarify.

Did you consider using a more comprehensive database for untargeted metabolomics e.g. MassBank?

Validity of the findings

In general the findings are well presented and valid.
There are some specific points that could be addressed:
Line 331-351: This paragraph is very convoluted and would benefit from editing to improve clarity.
Line 369-373. Please clarify what exact subset was used in this part of the study.
Why did the authors decide to focus only on the top 15 features to attempt to identify metabolites of interest that are separating the genera in question? How much % of the sample are these 15 metabolites representing? It seems like Pacifigorgia is not well represented in metabolites that are high in this specific genus. I believe it would be worth looking at more than 15 out of the thousands of metabolites found? Did you consider looking for metabolites that are only present in a certain genus? Maybe it could also be an option to look into the genera individually?
Maybe present a list of annotated features in your analysis?

Additional comments

Line 27: “in some cases”, please be more specific here.
Line 32-34. I would advise to swap the content of the sentence around to improve readability for the reader. “The contribution of metabolomic data into the integrative taxonomy of living organisms is still a matter of debate, due to uncertainties about the genetic and environmental origin of the metabolites.”
Overall I advise to thoroughly edit the first half of the abstract (until line 34) tailoring it more to the current project presented.

---

## Round 0.2 · Major Revisions

An expert reviewer who reevaluated your manuscript and their comments can be seen below along with an annotated manuscript in PDF. As you will see there are detailed comments particularly in terms of proper spelling, grammar and clarity that need to be addressed as well as other issues that were raised in the first round of revisions. Please ensure your rebuttlal letter clearly indicates what changes have been made and where or justify as to why not.

·

Basic reporting

The authors did their best to address all the raised issues from the first review of their manuscript, but are now encouraged to proof read thoroughly their manuscript as misspelling, convoluted sentences or sentences without verb were spotted.

Consequently, a few inconsistencies in the way data are represented can also lead to misunderstanding. In its current state, the manuscript still requires key revisions.

For examples:
• Line 281: ‘’additionally ‘’is misspelled
- Line 287: ‘’according with’’ should be replace by ‘’ according to’’
- Lines 299, 304: Euclidean is misspelled twice.
- Line 417: ‘’M344T129 are putatively’’, should be ‘’M344T129 were putatively”
- Line 470-471: the sentence lacks a verb.

- Line 471: ‘’the molecular ion produced by’’ should be replace by ‘’the molecular ion detected in’’. BE careful! A molecule is produced by an organism and a molecular ion is detected by mass spectrometry in the sample. Check also line 466

- Line 545: the sentence contains the word ‘’ tool’’ twice.
- Line 599-600: the last sentence should be revised what do the authors mean by chemical database? Is it spectral databases for dereplication?

The word ‘’approach’’ is intensely used and should ideally be replaced, whenever possible by a more precise and appropriate one. Examples
- Line 515: ‘’Using this integrative systematic approach’’ could be replaced by ’Using this integrative systematic method’’
- Line 546: ‘’we propose the inclusion of new tool Molecular Network approach tool’’ could be replaced by ‘’..we propose the inclusion of Molecular Network chemometrics to circumvent this key issue.”

- What do the authors mean by biochemical data? It is not really clear in the manuscript. Do they refer to MS-based metabolomic data? Or to the combination of DNA based analysis and metabolomics?

- Line 540: the authors keep repetition in the paragraph both metabolomic data and molecular data often in the same sentence and should be more precise in the wording to avoid any confusion to the reader. Molecular barcoding sequence could be used instead of molecular data for instance. Please check throughout.

- Line 544: the word annotation should be replaced by identification

- Line 545, in the discussion section, the authors mention the Molecular Networking as a NEW tool. But the methodology exists since 2016. As such, it can’t be considered as a new tool per se. It is now widely implemented in natural product chemistry for the structural dereplication of molecules through comparison of their MS2 spectra, notably with those available in the associated databases in the GNPS environment.

- Lines 550-551: the sentence should be revised. As written it is not a sentence a verb is missing.
- Line 552-555: The sentence should be split in two.

Experimental design

Nothing to add as previously reviewed the first time.


Raw data: The links to access to the raw data (DNA sequences and MS data) should be available through the manuscript unless there is a specific requirement from PeerJ.

Validity of the findings

All the required modifications or raised issues have been addressed in the revised version of the manuscript.
Nevertheless, some key clarifications need to be brought as per the putative identifications are concerned for data accuracy. I acknowledge the efforts made by the authors for putative identification of molecules but there are key mistakes in table 3 and in the corresponding paragraph starting in line 411 that need to be corrected.

When an MS-based putative identification is proposed, confidence levels should be added following the generally accepted guidance (see Schymanski et al. 2014, and more recently Alseekh et al. 2021)

- Lines 415-416: the authors stated: “the limited chemical studies … prevented from identifying the 13 molecular ions”. Nevertheless, a molecular formula without necessarily proposition a structure could possibly be given with a confidence level 4 (Schymanski et al. 2014).

- Line 420-421: ‘’by manual inspection of the mass spectra”: please precise HR-MS1 and MS/MS spectra?

Table 3: MS-based putative identification of 29 molecules from Octocoral species
- title ‘’molecular ion’’ is not correct. This title should ‘’measured m/z’’
- Title ‘’m/z’’ is not correct. This title should be “Ion Type” which should be presented [ M + H]+
- The authors should add two columns; one for the ‘’calculated m/z’’ and the other for the ‘’error in ppm’’.
- This information is crucial as it will reflect the accuracy in the proposed molecular formula leading to the putative identification.
- For example in the case of muricenone B: the calculated m/z for [M+H]+ = 333.2424 but the measured one is 333.1621 so the associated error is too high for the proposed molecular formula, leading to incorrect data. This issue concerns the entire table 3.

What VIP scores for these molecules? To which M..T.. feature these identifications pertained? Do they contribute to species discrimination? If not the authors should be really careful these data lack accuracy or worse are incorrect as presented.

Figure 4 legend
- M= Parent pass number, should be correct as M = Parent mass integer
- T= retention time number, should be corrected as T= retention time in min (? or in sec?)

Figure S4: The M..T code for the features is not used on the Y axis of the VIP scores. Please check the consistency of your codification throughout your manuscript.

Additional comments

There are still some key elements that required major revisions, although there is no additional data to provide.

I invite the authors to take their time to carefully proof read and double check their manuscript to reach more clarity in their statements, and avoid unnecessary repetitions.

Also one major issue pertain to MS based putative identification as already explained in the first review, and with a focus on table 3 in the second review.

---

## Round 0.3 · Minor Revisions

An expert reviewer has evaluated your revised manuscript and their comments can be seen below. As you will see this manuscript is acceptable for publication in PeerJ as soon as you correct the two minor suggestions made by the reviewer.

Reviewer 2 ·

Basic reporting

The manuscript has significantly improved since the first round of reviews. I have only 2 minor suggestions left.

Line 32) “unprecedented chemical structures” – I would use another word, unprecedented chemical structure sounds very vague and awkward.
Line 35-36) “our main objective was to assess the potential of metabolomics in the systematics of octocorals” Please rephrase this sentence, the potential of metabolomics to do what in the systematics? Improve them, verify them , disproof them?

Experimental design

The authors have managed to steer away from the case study towards a research paper that now fits within the scope of PeerJ.

Validity of the findings

The findings are valid and presented in a nice and coherent way.

Additional comments

No additional comments, The authors put a great deal of effort into revising the manuscript and came up with a nicely presented study.

---

## Round 0.4 · accepted · Accept

I am satisfied with the changes that have been made to the manuscript and find that it is ready for acceptance by PeerJ. Congratulations.